



# Global escalation of more frequent and intense compound heatwave-extreme precipitation events

Haoyu Jin [1, 2, 3, 9 *], Ke Zhang [1, 3, 4, 5, 6 *], Moyang Liu [7], Xuan Yu [8], Xu Yang [1, 4], Lijun Chao [1, 4], Pengfei Zhang [1, 4], Guoyan Liu [1, 4]

[1] State Key Laboratory of Water Disaster Prevention, Hohai University, Nanjing, Jiangsu, 210024, China

[2] Key Laboratory of Transportation Meteorology of China Meteorological Administration, Nanjing Joint Institute for Atmospheric Sciences, Nanjing 210041, China

[3] Yangtze Institute for Conservation and Development, Nanjing, Jiangsu, 210024, China

[4] College of Hydrology and Water Resources, Hohai University, Nanjing, Jiangsu, 210024, China

[5] China Meteorological Administration Hydro-Meteorology Key Laboratory, Hohai University, Nanjing, Jiangsu, 210024, China

[6] Key Laboratory of Water Big Data Technology of Ministry of Water Resources, Hohai University, Nanjing, Jiangsu, 210024, China

[7] The Fenner School of Environment and Society, The Australian National University (ANU), Canberra, ACT 0200, Australia

[8] State Key Laboratory of Soil and Sustainable Agriculture, Institute of Soil Science, Chinese Academy of Sciences, Nanjing 210008, China

[9] Meteorological Administration Hydro-Meteorology Key Laboratory, China

* **Correspondence:** Haoyu Jin (haoyu.jin@hhu.edu.cn) and Ke Zhang (kzhang@hhu.edu.cn)

**Abstract.** Compound heatwave-extreme precipitation (CHWEP) events, the rapid succession of heatwaves and extreme precipitation, pose growing compound and cascading risks. However, global-scale comparisons of their spatiotemporal evolution against single extremes remain limited. This study systematically examines the changes in CHWEP and corresponding single extremes from 1980 to 2100 using climate observations and projections under SSP (Shared Socioeconomic



Pathway) 2-4.5 and SSP5-8.5 scenarios. We find that CHWEP exhibit higher frequency, stronger precipitation, and longer heatwave duration in mid-to-high latitudes, while tropical CHWEP feature more intense heatwaves than single heatwave events. These spatial contrasts persist in future projections. Under both scenarios, CHWEP and single extreme metrics intensify globally by 2056-2100, with post-heatwave precipitation exceeding that of single precipitation extremes, particularly under SSP5-8.5, highlighting sensitivity to greenhouse forcing. Critically, the co-occurrence is non-random, indicating an emerging physical linkage. In the tropics, the likelihood of extreme rainfall following heatwaves increases markedly. Our findings demonstrate that CHWEPs are evolving into a distinct, intensifying hazard class, necessitating their integration into climate resilience, early warning, and adaptation frameworks.

## 1 Introduction

Global warming has significantly increased the frequency and intensity of extreme climate events, posing growing threats to human societies and ecosystems (Diffenbaugh et al., 2017; Gu et al., 2022; Myhre et al., 2019). Among these, heatwaves and extreme precipitation are two of the most widespread and hazardous types of extreme weather (Raghavendra et al., 2018; Sun et al., 2024). They are typically treated as independent phenomena, as their physical mechanisms and manifestations are contrasting, and they rarely occur simultaneously at the same location (Sauter et al., 2023a; Sun et al., 2023). However, a non-synchronous but causally linked lagged response may exist between them. For instance, prolonged heatwaves can alter land-atmosphere interactions, such as inducing soil drying, enhancing boundary layer warming, and accumulating convective instability energy, thereby creating favorable conditions for subsequent extreme precipitation, forming a "heat-then-rain" compound event chain (Chen et al., 2022; Li et al., 2025; Liu et al., 2024; Yang and Yuan, 2025).



The concept of compound extreme events was first introduced in the IPCC (Intergovernmental Panel on Climate Change) Fifth Assessment Report (AR) and has been further

refined in AR6 into four categories: multivariate, multi-event, temporally compounding, and spatially compounding extremes (Bevacqua et al., 2023; Fang et al., 2025; Fischer and Knutti, 2015). Studies have shown that the impacts of compound events often far exceed the simple sum of individual extremes, exhibiting significant nonlinear amplification effects (Ning et al., 2022; Zhao et al., 2022). Among these, compound heatwave-extreme precipitation (CHWEP) events

have attracted increasing attention due to their potential to trigger "drought-to-flood" transition disaster chains (Ganguli and Merz, 2024; Götte and Brunner, 2024; Sauter et al., 2023b; Zhou et al., 2023). However, current research is largely limited to regional case studies, and the global spatiotemporal patterns, trends, and underlying drivers of CHWEP events remain poorly understood, especially in comparison with single extreme events (Jin et al., 2024; Mazdiyasni and

AghaKouchak, 2015; Miao et al., 2024). More critically, the future changes in the occurrence probability, intensity, and associated compound risks of CHWEP events under a warming climate remain largely unexplored and constitute a major knowledge gap (Séférian et al., 2019; Xiong et al., 2023; You and Wang, 2021; Zscheischler et al., 2018).

In this study, based on multiple reanalysis datasets and the multi-model ensemble from the

Coupled Model Intercomparison Project Phase 6 (CMIP6), we define CHWEP events along with their corresponding single heatwave and extreme precipitation events. First, we systematically compare the historical changes (1980-2024) and future projections (2056-2100) in the frequency, intensity, and duration of CHWEP events and their single-component counterparts at the global scale. Second, we investigate the potential linkage between heatwaves and extreme precipitation

events through probabilistic analysis, demonstrating that CHWEP events are not merely the result



of random co-occurrence. To our knowledge, this is the first comprehensive, global-scale, multi-model assessment of CHWEP events across the full process, from past detection to future projection. This research provides critical scientific support for extreme weather early warning systems and holds significant practical implications for enhancing societal climate resilience.

In the remainder of this manuscript, Section 2 describes the three reanalysis datasets and four CMIP6 global climate models (GCMs) used to identify CHWEP events. Section 3 presents the methodologies for defining heatwave, extreme precipitation, and CHWEP events, and further examines the statistical significance of differences among event types as well as the randomness of their co-occurrence. Section 4 presents the results along with interpretability analyses. Section
5 discusses key insights derived from this study. Finally, Section 6 provides the concluding remarks.

## 2  Data

In    this    study,    we    selected    three    global    reanalysis    datasets:    ERA5 (https://cds.climate.copernicus.eu/datasets/derived-era5-land-daily-statistics?tab=download), MERRA-2
(https://disc.gsfc.nasa.gov/datasets/M2SDNXSLV_5.12.4/summary?keywords=merra2),    and JRA-55 (https://jra.kishou.go.jp/JRA-55/index_en.html), which are widely used in climate research, weather forecasting, hydrological modeling, and extreme event analysis (Chen et al., 2019; Huang et al., 2015; Wang et al., 2019a). Daily precipitation and daily maximum temperature data from 1980 to 2024 were used. Due to differences in spatial resolution and performance among
the datasets, we resampled all data to a uniform $1° \times 1°$ grid using kriging interpolation and adopted the ensemble mean as the final observational reference. To analyze the future evolution of CHWEP events, we used daily precipitation and daily maximum temperature outputs from four CMIP6 GCMs for the period 2056-2100, under two Shared Socioeconomic Pathway (SSP) scenarios:



SSP2-4.5 (representing a medium emission scenario) and SSP5-8.5 (representing a high emission

scenario) (Table S1) (https://aims2.llnl.gov/search/cmip6/) (Balaji et al., 2018; Juckes et al., 2020).

Similarly, the model data were interpolated to a 1° × 1° grid using kriging, and the multi-model

ensemble mean was computed.

Given the systematic biases between climate model simulations and reanalysis data, we

applied a quantile mapping method to perform simple bias correction on the CMIP6 model outputs

(Xu et al., 2021; Xu and Han, 2019). However, since this study primarily focuses on the relative

differences between CHWEP events and single extreme events, rather focus on absolute

magnitudes, the precise accuracy of the absolute values does not affect the overall conclusions.

## 3 Methods

### 3.1 Identification and characteristics of CHWEP events

In this study, we first define heatwave events and extreme precipitation events. A heatwave is

identified as a period of at least three consecutive days (≥3 days) during which the daily maximum

temperature exceeds the 90th percentile threshold of historical daily maximum temperatures

(Wang et al., 2019b; Zscheischler et al., 2020). Extreme precipitation is defined as a daily

precipitation amount exceeding the 90th percentile threshold derived from all wet days (>1 mm)

in the historical period. The threshold for future periods is determined based on the threshold of

historical periods for extreme events. The CHWEP event is characterized by the occurrence of an

extreme precipitation event within 7 days following a heatwave event. This 7-day window is

chosen to ensure a physically plausible linkage between the two events while maintaining a

sufficient sample size for statistical analysis (Ridder et al., 2020; Sauter et al., 2023b). Events that

do not meet the CHWEP criteria are classified as single heatwave or single extreme precipitation

events (Fig. 1). We define four metrics to characterize the properties of CHWEP and single extreme





events: Frequency, Heatwave intensity, Extreme precipitation intensity, Duration. The calculation

formulas are as follows:

$$F_C = \sum_i CHWEP \tag{1}$$

$$F_S = \sum_i HW \tag{2}$$

$$IT_C = mean(\sum_j T_{CHWEP}) \tag{3}$$

$$IT_S = mean(\sum_j T_S) \tag{4}$$

$$IP_C = mean(\sum_j P_{CHWEP}) \tag{5}$$

$$IP_S = mean(\sum_j P_S) \tag{6}$$

$$D_C = mean(\sum_i D_{CHWEP}) \tag{7}$$

$$D_S = mean(\sum_i D_S) \tag{8}$$

where $F$ denotes the frequency of events, $IT$ represents the mean daily maximum temperature

during the extreme event, $IP$ denotes the mean precipitation amount during the extreme

precipitation event, and $D$ refers to the duration of the heatwave in days, subscript $C$ indicates

CHWEP events, subscript $S$ denotes single extreme events, $i$ denotes the event index, and $j$

denotes the day within a given event.



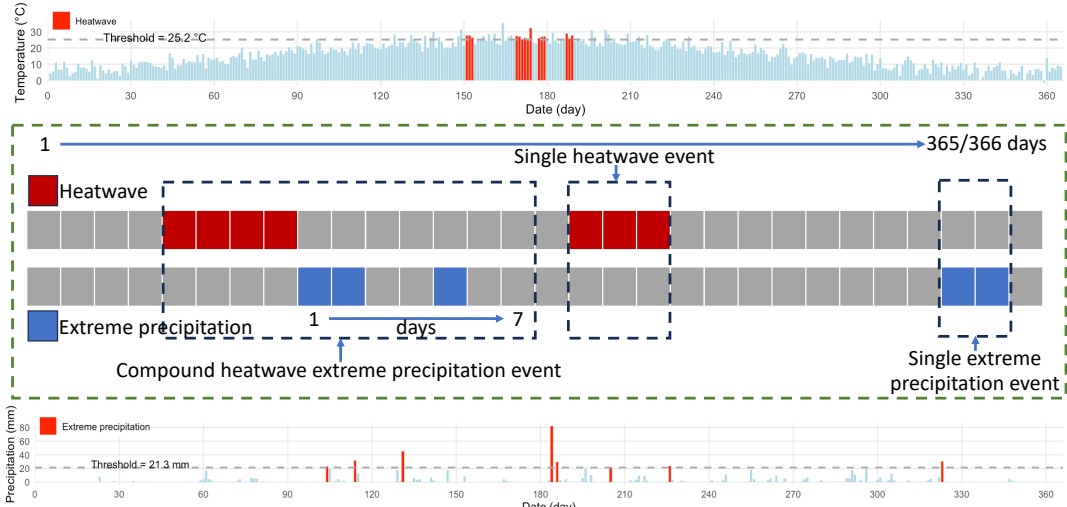

**Figure 1.** Identification diagram of CHWEP event.

## 3.2 Significance test of difference

In this study, the Wilcoxon rank-sum test (WRST) is employed to assess the statistical differences in extreme event characteristics between CHWEP events and single extreme events, as well as between future projection periods and the historical baseline period. The WRST is a non-parametric statistical method used to determine whether there is a significant difference in the medians of two independent samples (Guerreiro et al., 2018). It does not assume any specific data distribution (e.g., normality), making it particularly robust for datasets with skewed distributions, small sample sizes, or the presence of outliers. This method is therefore well suited for analyzing extreme climate events, which often exhibit non-normal behavior and high variability.

## 3.3 Event coincidence analysis

CHWEP events may arise either from the random co-occurrence of heatwaves and extreme precipitation, or through a potential physical linkage whereby the occurrence of heatwaves promotes the subsequent development of extreme precipitation (Liu et al., 2024; Ning et al., 2022). To investigate the underlying mechanisms of CHWEP events, we employ Event Coincidence





Analysis (ECA). ECA is a statistical method designed to assess the potential associations between discrete events in two or more time series (Donges et al., 2016; Xiao et al., 2025). It is particularly effective in detecting whether events in different systems occur simultaneously or in close temporal proximity more often than expected by chance, thereby helping to identify possible causal relationships or shared driving factors (Sun et al., 2024). The method quantifies the strength of such linkages by comparing the observed coincidence rate with the expected coincidence rate under randomness. The corresponding formulas are as follows:

$$P_{obs} = C_{AB}/N_A \qquad (9)$$

$$P_{rand} = N_B \times \Delta_t/T \qquad (10)$$

where $P_{obs}$ denotes the observed coincidence probability, $C_{AB}$ represents the number of extreme precipitation events occurring within a time window of $\Delta_t$ (set to 7 days in this study) following each heatwave event, $N_A$ is the total number of heatwave events. $P_{rand}$ is the expected coincidence probability under the null hypothesis of randomness, $N_B$ is the total number of extreme precipitation events, and $T$ is the total length of the observational period, in this case, the number of days in a year (365 or 366 for leap years).

## 4 Results

### 4.1 Historical comparison of CHWEP and single extreme events

CHWEPs exhibit distinct spatial frequency patterns compared to single heatwaves. CHWEPs occur more frequently in mid-to-high latitudes (Fig. 2a), whereas single heatwaves are more prevalent in tropical regions and the Middle East (Fig. 2b). The spatial patterns of heatwave intensity are broadly similar between CHWEPs (Fig. 2c) and single heatwaves (Fig. 2d), with hotspots in the Sahara Desert, the Middle East, and Australia. Similarly, the intensity patterns of



extreme precipitation during CHWEPs (Fig. 2e) resemble those of single extreme precipitation

events (Fig. 2f), with high-intensity regions located in the equatorial zone, northern Argentina, and

southern China. The spatial distribution of heatwave duration also shows strong similarity between

CHWEPs (Fig. 2g) and single heatwaves (Fig. 2h), with prolonged durations predominantly

observed in the equatorial regions, the Middle East, and India.










**Figure 2.** Spatial distribution of the mean values of the frequency (a and b), heatwave intensity (c and d), extreme precipitation intensity (e and f), and heatwave duration (e and f) of CHWEP (a, c, e, and g) and single extreme events (b, d, f, and h) from 1980 to 2024.

CHWEPs occur more frequently than single heatwaves in mid-to-high latitude regions (Fig. 3a). The heatwave intensity associated with CHWEPs is significantly higher than that of single heatwaves in tropical regions (Fig. 3b). Across most mid-to-high latitude areas, the extreme precipitation intensity during CHWEPs exceeds that of single extreme precipitation events (Fig. 3c). Moreover, the duration of heatwaves within CHWEPs is notably longer than that of single heatwaves in the mid-to-high latitudes (Fig. 3d). These results indicate that the impacts of

CHWEPs are more pronounced in mid-to-high latitude regions compared to low latitudes.

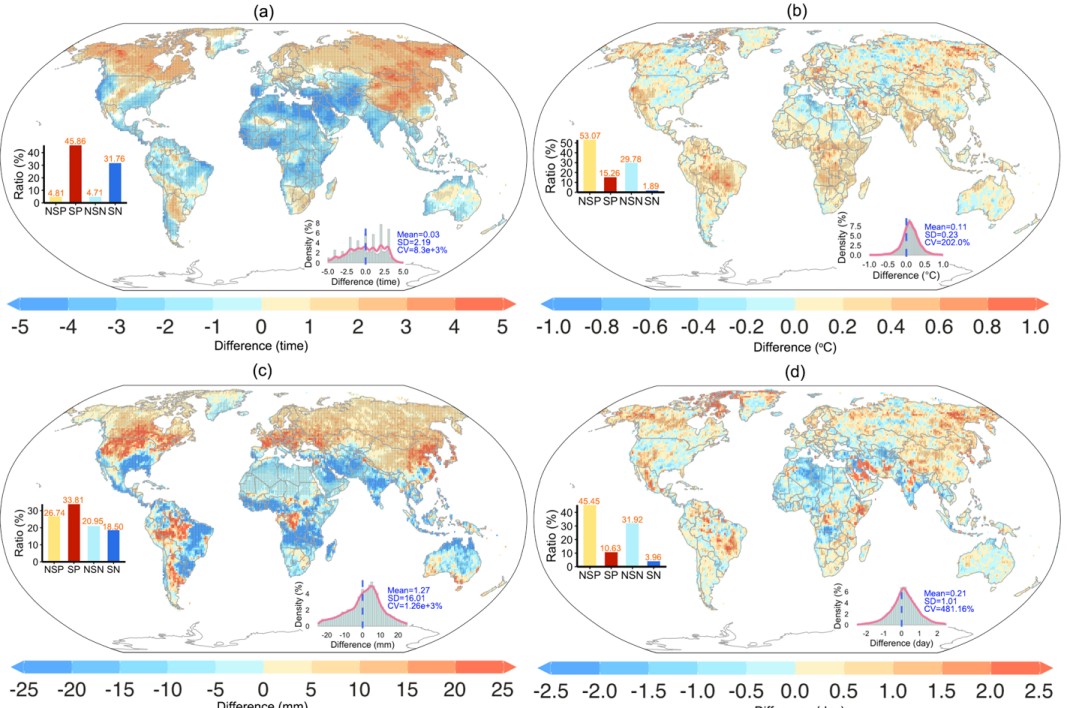

**Figure 3.** Spatial distribution of median differences between CHWEP and single extreme events in frequency (a), heatwave intensity (b), extreme precipitation intensity (c), and heatwave





duration (d) from 1980 to 2024. (Note: In the left bar chart, NSP denotes Non-Significant

Positive, SP denotes Significant Positive, NSN denotes Non-Significant Negative, and SN

denotes Significant Negative. The same abbreviations apply hereinafter.)

## 4.2 Future changes and comparisons of CHWEP and single extreme events

Under the SSP2-4.5 (Fig. S1) and SSP5-8.5 (Fig. S2) scenarios, the spatial patterns of

CHWEPs and single extreme events are broadly similar to those observed during the historical

period (Fig. 2), indicating a consistent spatial organization of extreme events across time periods.

The spatial distribution of differences in event characteristics between CHWEPs and single

extremes also remains largely unchanged under future projections. Specifically, the frequency of

CHWEPs (Fig. 4a), the intensity of associated extreme precipitation (Fig. 4e), and heatwave

duration (Fig. 4g) are substantially higher in mid-to-high latitudes compared to single events (Fig.

4b, f, and h). Conversely, heatwave intensity during CHWEPs is greater in tropical regions than

that of single heatwaves (Fig. 4c and d). Notably, the magnitude of these differences is more

pronounced under the SSP5-8.5 scenario than under SSP2-4.5, suggesting an amplification of

compound event characteristics under stronger warming.







**Figure 4.** Spatial distribution of median differences between CHWEP and single extreme events in frequency (a and b), heatwave intensity (c and d), extreme precipitation intensity (e and f), and heatwave duration (g and h) from 2056 to 2100 under SSP2-4.5 (a, c, e, and g) and SSP5-8.5 (b, d, f, and h) scenarios.

Globally averaged trends in CHWEPs and single extreme events show that the frequency (Fig. 5a), heatwave intensity (Fig. 5b), extreme precipitation intensity (Fig. 5c), and heatwave duration (Fig. 5d) of both compound and single events are increasing over time. Notably, the rates of increase in frequency and heatwave duration are faster than those of heatwave intensity and extreme precipitation intensity. Furthermore, under the SSP5-8.5 scenario, all extreme event metrics are substantially higher than under the SSP2-4.5 scenario, indicating a strong dependence on the level of future warming.

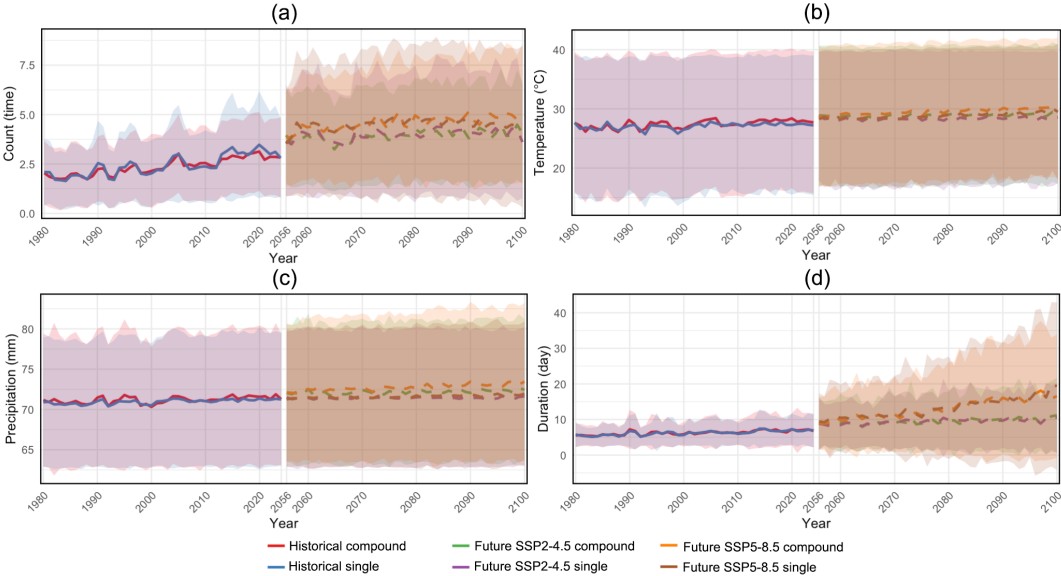

**Figure 5.** Changes in the frequency (a), heatwave intensity (b), extreme precipitation intensity (c), and heatwave duration (d) during the historical and future periods of CHWEP and single extreme events.



Compared to the historical period, both CHWEPs and single extreme events show increases in frequency, heatwave intensity, extreme precipitation intensity, and heatwave duration over more than half of the global land area under the SSP2-4.5 (Fig. S3) and SSP5-8.5 (Fig. S4) scenarios. The magnitude of these increases is substantially larger under the SSP5-8.5 scenario, indicating a strong dependence on the level of global warming.

Under the SSP2-4.5 and SSP5-8.5 scenarios, the proportion of grid cells showing positive changes in attribute metrics of CHWEP and single extreme events is overwhelmingly dominant. Under SSP5-8.5, the frequency (Fig. 6a), heatwave intensity (Fig. 6b), and heatwave duration (Fig. 6d) of both CHWEP and single extreme events are higher than under SSP2-4.5, with similar magnitudes of change. Furthermore, we find that under both scenarios, the extreme precipitation

intensity associated with CHWEP is significantly greater than that of single extreme precipitation events (Fig. 6c). Notably, under SSP2-4.5, the precipitation intensity in CHWEP even exceeds that of single extreme events under the more severe SSP5-8.5 scenario. This indicates that CHWEP events feature stronger extreme precipitation, and heatwaves tend to amplify the intensity of concurrent rainfall extremes.

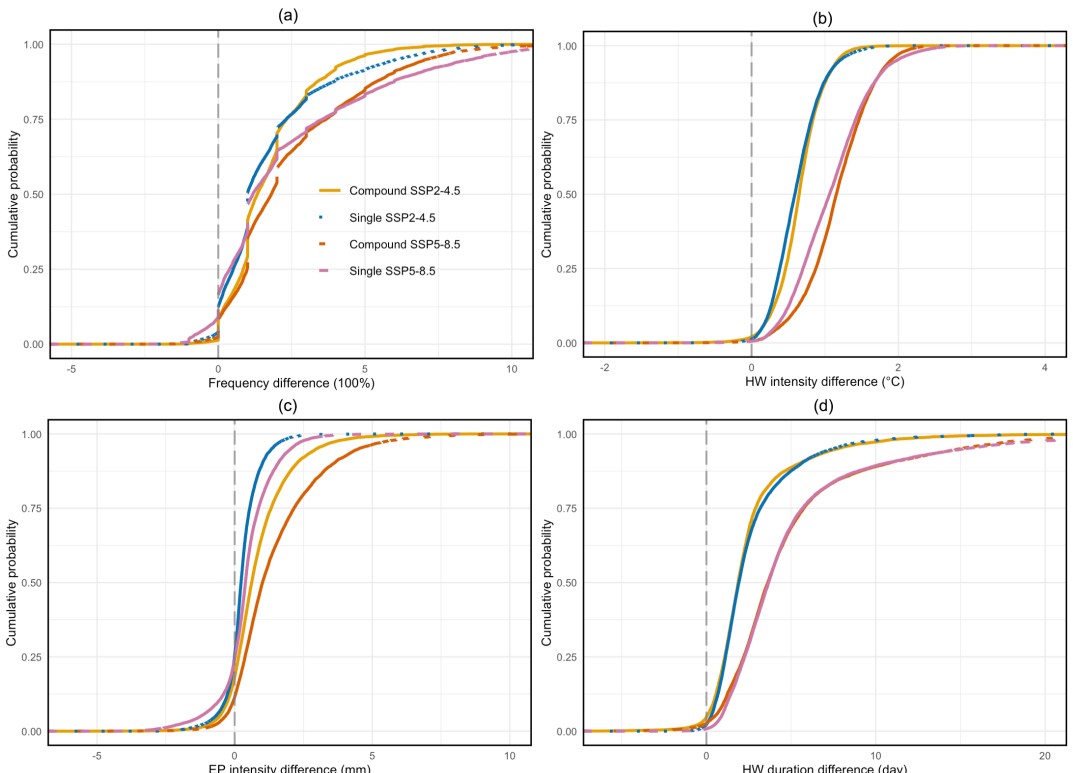


**Figure 6.** Comparison of the cumulative probability of the median differences between the frequency (a), heatwave intensity (b), extreme precipitation intensity (c), and heatwave duration (d) of CHWEP and single extreme events under the SSP2-4.5 and SSP5-8.5 scenarios and the historical period.

### 4.3 Probability of encountering heatwaves and extreme precipitation events


The observed probability of co-occurring heatwaves and extreme precipitation is higher in mid-to-high latitude regions (Fig. 7a), with smaller standard deviations (SD, indicating lower variability) (Fig. 7d). In contrast, the random probability of their co-occurrence is higher in tropical regions (Fig. 7b), but with greater variability (Fig. 7e). The ratio of observed to random probability exceeds 1 over more than 66% of the globe (Fig. 7c), particularly in mid-to-high latitudes, where




the variability is relatively low (Fig. 7f), suggesting that heatwave events in these regions are more likely to be followed by extreme precipitation. In terms of trends, the observed probability (Fig. 7g), random probability (Fig. 7h), and their ratio (Fig. 7i) are increasing more rapidly in tropical regions, indicating a growing likelihood of CHWEP in the tropics in the future.

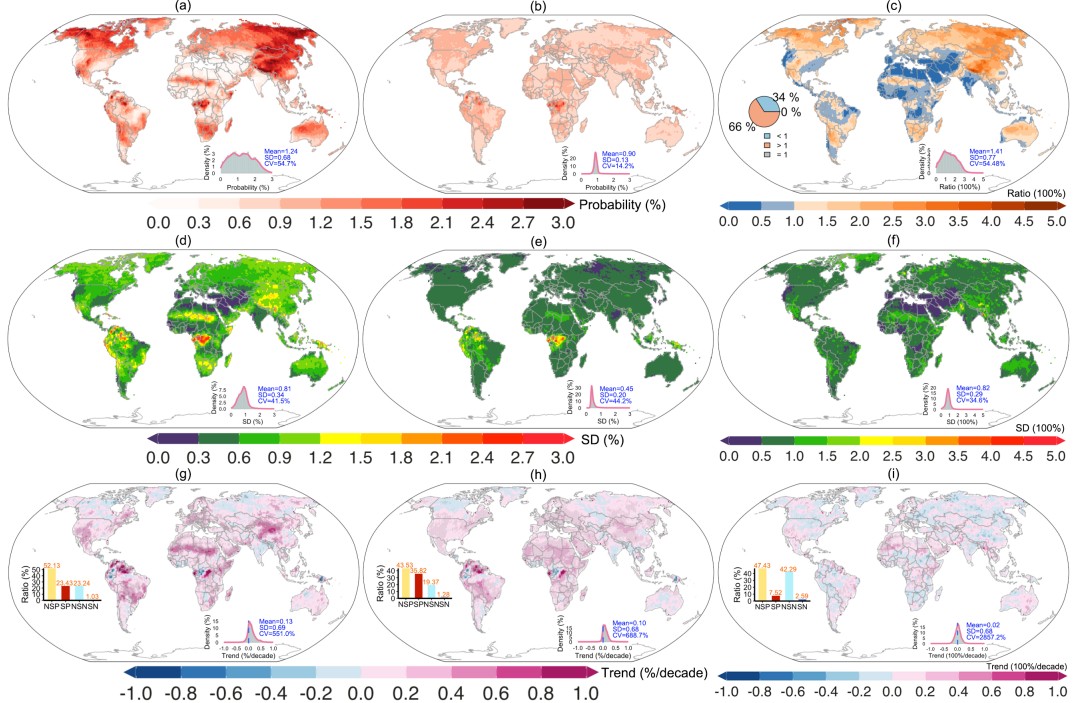

**Figure 7.** Spatial distributions of the mean observed probability (a), mean random probability (b), and mean ratio of observed probability to random probability over the period 1980-2024 in the first row; the second row shows the spatial distributions of their standard deviations (d-f); the third row presents their spatial patterns of linear trends (g-i).

In the future period, the observed probability of co-occurring heatwaves and extreme precipitation remains higher in mid-to-high latitude regions (Fig. 8a and d), while their random co-occurrence probability continues to be higher in tropical regions (Fig. 8b and e). Under the SSP5-8.5 scenario, both observed and random co-occurrence probabilities are notably higher than





under the SSP2-4.5 scenario. Under SSP2-4.5, 70.9% of the globe shows a ratio of observed to

random co-occurrence probability greater than 1 (Fig. 8c), increasing to 76.2% under SSP5-8.5

(Fig. 8f), indicating an intrinsic linkage mechanism between heatwaves and extreme precipitation.

This suggests that, in the future, a growing number of regions worldwide will experience extreme

precipitation following heatwave events.

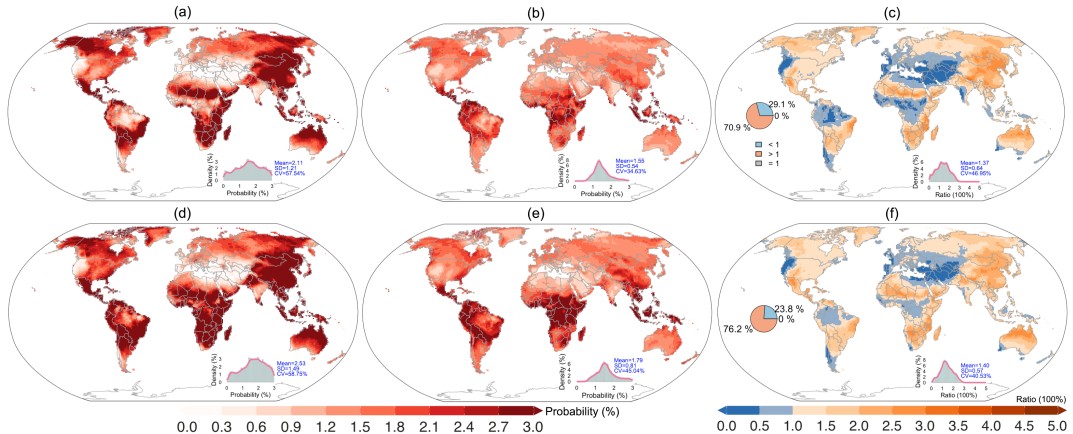

**Figure 8.** Spatial distributions of the mean observed probability (a and d), mean random

probability (b and e), and mean ratio of observed probability to random probability (c and f)

under the SSP2-4.5 (a-c) and SSP5-8.5 (d-f) scenarios for the period 2056-2100.

Globally, both the observed and random probabilities of heatwave and extreme precipitation

co-occurrence show increasing trends from the historical period to the future period (Fig. 9a), with

higher values and faster growth under the SSP5-8.5 scenario. This indicates that more CHWEP

will occur globally under future climate change. Although both observed and random co-

occurrence probabilities are rising, their ratio shows no significant trend over time, however, the

ratio remains consistently greater than 1 (Fig. 9b). This demonstrates that CHWEP events are not

merely the result of random coincidence, heatwaves tend to promote the subsequent occurrence of

extreme precipitation, suggesting a physical linkage between the two extremes.



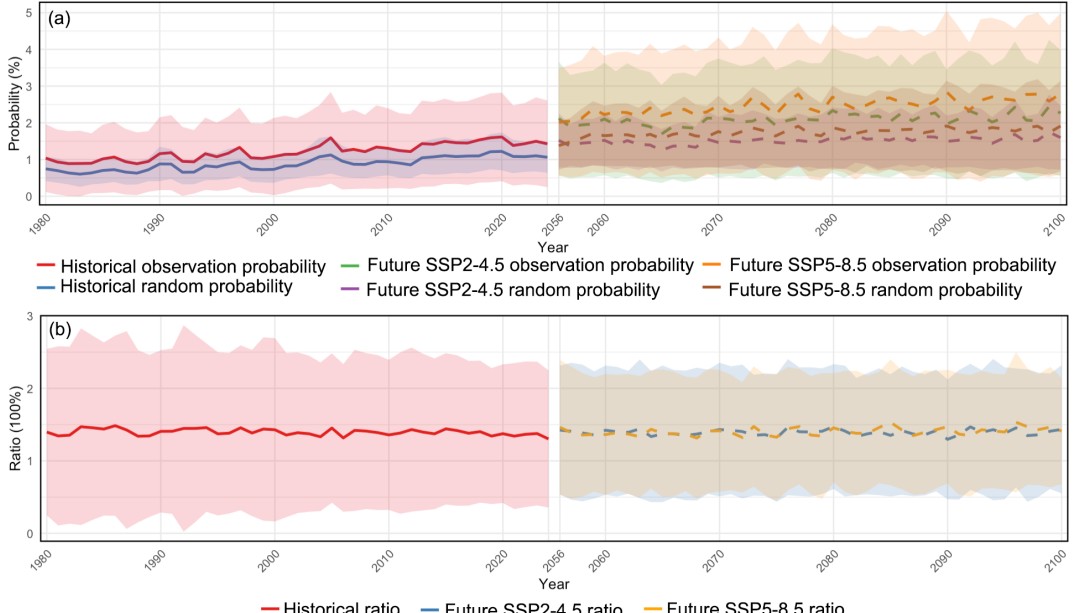

**Figure 9.** Changes in the observed probability and random probability (a), and their ratio (b) for the global co-occurrence of heatwaves and extreme precipitation between the historical period (1980-2024) and the future period (2056-2100).

Under future scenarios, the observed (Fig. 10a and d) and random (Fig. 10b and e) probabilities of heatwave and extreme precipitation co-occurrence increase over most of the globe, with larger increases in tropical regions. The magnitude of change in co-occurrence probability is greater under the SSP5-8.5 scenario than under SSP2-4.5. The ratio of observed to random co-occurrence probability (Fig. 10c and f) increases more in tropical regions but decreases in mid-to-

high latitude regions, with larger changes under SSP5-8.5. This indicates that CHWEP events will become more frequent in the future, and the likelihood of extreme precipitation following heatwaves will increase more substantially in the tropics.



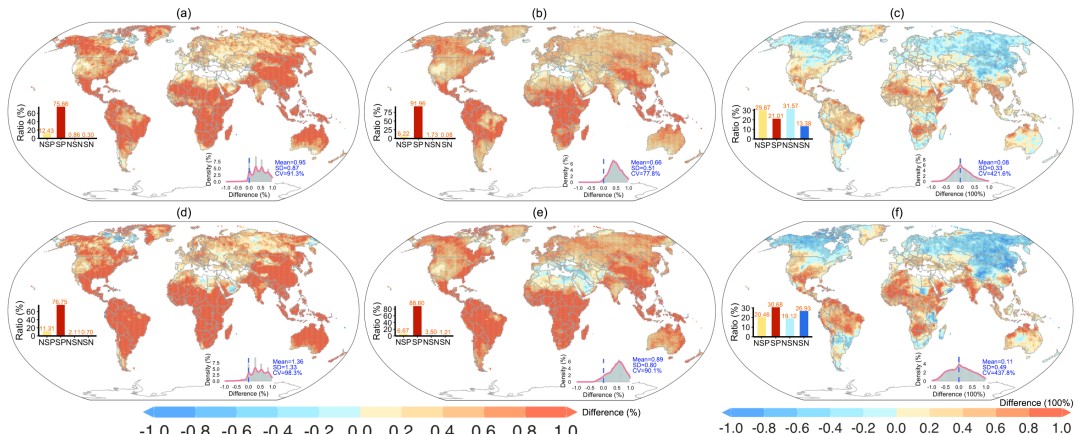

**Figure 10.** Spatial distribution of the differences in median values of observed probability (a and d), random probability (b and e), and their ratio (c and f) for the co-occurrence of heatwaves and extreme precipitation between the future period (under SSP2-4.5 (a-c) and SSP5-8.5 (d-f) scenarios) and the historical period.

## 5 Discussion

Compound extreme events are receiving increasing attention from both the scientific community and society, as regions experiencing concurrent or sequential extremes often face compounded or cascading impacts that lead to more severe socioeconomic and ecological damage than single extremes alone (Götte and Brunner, 2024; Sauter et al., 2023a; Sun et al., 2023). This study focuses on CHWEP, revealing that beyond the cumulative impact of sequential extremes, the integrated intensity of compound events exceeds that of individual extremes across most regions, resulting in heightened risks (Jin et al., 2024; Xiao et al., 2025). Previous studies suggest that the choice of extreme event thresholds and the time window for defining compound events have limited influence on the identification of their underlying linkages, whereas sufficient sample sizes enable more robust detection of long-term trends (Bevacqua et al., 2021; Diffenbaugh et al.,



2017; Fang et al., 2025). Although climate models may exhibit biases in simulating absolute
magnitudes, this study emphasizes relative changes, focusing on how compound events evolve
rather than their exact intensity, ensuring the robustness of the findings (Chen et al., 2022; Ganguli
and Merz, 2024; Lei et al., 2024).

This study shows that both CHWEP and single extreme events, such as single heatwaves or
extreme precipitation, are increasing significantly in frequency, intensity, and duration, indicating
that most regions worldwide are shifting toward a climate regime characterized by more frequent
and intense extremes (Ning et al., 2022; Sun et al., 2024; Xiao et al., 2025). Crucially, the
intensification of CHWEPs not only implies a higher risk of concurrent disasters but also reveals
a dangerous "rapid transition" phenomenon, namely, a swift shift from extreme heat to extreme
rainfall (Ning et al., 2022; Ridder et al., 2020; Xie and Zhou, 2023). This abrupt transformation
from drought-like heatwaves to flash flooding poses a severe challenge to traditional disaster
preparedness systems, which are typically designed to address single, single-hazard events (Kumar
et al., 2024; You and Wang, 2021; Zhou et al., 2023).

Furthermore, the spatiotemporal evolution patterns of compound and single extreme events
revealed in this study provide critical scientific foundations for policymakers, urban planners, and
emergency management agencies (Bevacqua et al., 2023; Gu et al., 2022; Myhre et al., 2019). By
identifying high-risk regions and key temporal windows, relevant authorities can develop more
resilient integrated early warning systems (Almeida et al., 2016; Dike et al., 2022; Sauter et al.,
2023b). Advancing predictive models capable of capturing the transition from heatwaves to heavy
rainfall is essential for effective water resource management, agricultural planning, public health
response, and infrastructure safety (Kumar et al., 2024; Sun et al., 2023). In the context of climate
change, where the synergistic and cascading nature of extreme events is intensifying, enhancing



societal resilience and long-term sustainability in the face of compound climate disasters has become increasingly urgent and necessary (Miao et al., 2024; Sauter et al., 2023a; Woolway et al., 2021).

## 6 Conclusions


This study systematically analyzes the spatiotemporal evolution patterns of CHWEP and single extreme events. Results show that in mid-to-high latitude regions, CHWEPs exhibit higher frequency, greater extreme precipitation intensity, and longer heatwave duration compared to single extremes. In tropical regions, the heatwave intensity within CHWEPs exceeds that of single

heatwaves. These spatial contrasts persist under both future SSP2-4.5 and SSP5-8.5 scenarios. Compared to the historical period, metrics of both CHWEPs and single extremes increase across most of the globe in the future, with extreme precipitation following heatwaves becoming notably more intense than single extreme precipitation events. Under the high-emission SSP5-8.5 scenario, the intensification of extremes is greater than under the moderate-emission SSP2-4.5 scenario. The

co-occurrence of extreme precipitation after heatwaves is not merely a result of random coincidence. In the future, the likelihood of extreme precipitation following heatwaves will increase, particularly in tropical regions.

*Code availability.* All R codes can be provided by the corresponding authors upon request.

*Data availability.* The three global reanalysis datasets: ERA5

(https://cds.climate.copernicus.eu/datasets/derived-era5-land-daily-statistics?tab=download),

MERRA-2

(https://disc.gsfc.nasa.gov/datasets/M2SDNXSLV_5.12.4/summary?keywords=merra2), and

JRA-55 (https://jra.kishou.go.jp/JRA-55/index_en.html) can be accessed at the links. The four



CMIP6 GCMs for the period 2056-2100, under two Shared Socioeconomic Pathway (SSP)

scenarios: SSP2-4.5 and SSP5-8.5 can be accessed at https://aims2.llnl.gov/search/cmip6/.

*Author contributions.* KZ and HJ acquired the supervision and funding for this research. All authors contributed to the study conception, design, and methodology. ML, XY, and XY undertook the tasks of programming, data collection, result derivation, and interpretation. The original draft was prepared by HJ and ML and subsequently revised by LC, PZ and GL.

*Competing interests.* The contact author has declared that neither of the authors has any competing interests.

*Acknowledgements.* Data processing and visualization were performed with R (R Core Team, 2024). The authors express their gratitude to the reviewers.

*Financial support.* This study was supported by National Key Research and Development Program

of China (2023YFC3006505), the Special Fund Project of Jiangsu Province Science and Technology Program (BZ2024035), the fund of National Key Laboratory of Water Disaster Prevention (524015222), the Project "Applied Scientific Research on the 'Three-Line Defense' Strengthening Foundation Project for Rainfall and Water Monitoring & Forecasting in Shandong Province" (37000000025001720240235), the Open Foundation of Beijige (BJG202513), and the

Open Foundation of China Meteorological Administration Hydro-Meteorology Key Laboratory (Grant No. 24SWQXZ055).

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
