# Peer review of "Global escalation of more frequent and intense compound heatwave-extreme precipitation events"

_EGUsphere, 2025_

## Author Comment (AC1)

* * *
**Response to the RC1**

Dear Reviewer,

We sincerely thank the reviewer for your care, support, and valuable assistance with our research. Your insightful comments and suggestions have been of tremendous help in continuously improving and enhancing both the content and overall quality of this study. Upon receiving your review comments, we promptly revised the manuscript in accordance with your recommendations, striving to fully meet all requested revisions. The revised portions have been highlighted in red in the manuscript for your convenient review. Our point-by-point responses to your comments are provided below.

Thank you again for your time and kind assistance.

Best regards,
Haoyu Jin

The authors present an analysis of compound heatwave extreme precipitation (CHWEP) events using statistics computed an ensemble of 3 reanalysis data (in the past) and 4 CMIP models (for projection). They first show the differences on several statistics between CHWEP and single heat waves (SH). Then, they show how these statistics vary in the future, under several SSPs.

Overall, I found that this analysis suffers from several methodological issues that are detailed below. The paper is not very well written, with important quantities not properly defined, equations I could make sense of, etc. In my opinion, this work cannot be accepted for publication in HESS, unless completely reworked.

Main comments:

1. Currently, the authors compute the statistics in the past on reanalysis and in the future on climate models. It is well known that climate models are highly biased if not corrected with bias correction methods (see eg François et al., 2020). Is it the case here?

If the answer is 'no', then I don't see how it is possible to draw any definitive conclusions when comparing the future (using CMIP models) and the past (with reanalysis) as done in Sections 4.2 and 4.3 (Figures 6 - 10 and S3 - S4). If the answer is 'yes', the authors should nevertheless first check that the WRT does not reject the equality of the distribution when the statistics are computed on the models during the past periods. This is a preliminary study that is absolutely necessary. Otherwise, one does not know for sure whether the differences observed are due to climate change or to differences between models and reanalysis.

*Response:* The authors applied quantile delta mapping (QDM) to bias-correct precipitation and temperature data from CMIP6 models under different future scenarios. QDM is an advanced bias correction method that preserves the climate change signal in future projections. It improves upon

traditional quantile mapping (QM) by not directly mapping future simulations onto the historical observational distribution. Instead, it transfers the modeled distributional changes between the historical and future periods, i.e., the "delta" or incremental shift, onto the observational distribution, thereby avoiding the suppression of climate change trends inherent in CMIP model projections. The Wilcoxon rank-sum test (WRST) is a non-parametric hypothesis test used to assess whether two independent samples originate from the same population (or populations with identical medians). It does not assume normality of the data and is therefore widely employed when the assumptions of parametric tests (such as the t-test) are violated. In this study, we use WRST to statistically evaluate the median differences in precipitation and temperature characteristics between compound heatwave-extreme precipitation (CHWEP) events and single extreme events (extreme precipitation and heatwaves), thereby quantitatively distinguishing CHWEP events from individual extremes. We thank the reviewer for pointing out this issue, which helps the authors further improve and refine this study.

2. Equations (1) - (8) are unclear. For example, in Eq. (1), there is a sum indexing over $i$, but I don't understand what the index is exactly (and there is no $i$ in CHWEP). Also I don't understand what we are summing exactly. Eq. (3) is even less clear. What is actually computed? The mean (over all events) of the sum of the temperature (withing each events)? In this case, one should see numbers above 100? I am really lost here.

The same kind of criticism could be made for each equation. Note also that these statistics are computed on a 45 year period, across several members of an ensemble. I guess there is some sort of averaging over the years that should be made apparent.

Equation (10) is also very unclear). Why is $P_{rand}$ a coincidence probability?

*Response:* The authors have revised Sections 3.2 and 3.4 to provide a clearer description of the computation of attributes for CHWEP and single extreme events, as well as the observed and random coincidence probabilities for CHWEP events. Figure 1 has also been redrawn to more clearly illustrate the extracted attribute metrics. Specifically, we first identify extreme precipitation and heatwave events using precipitation and temperature thresholds, respectively. CHWEP events are then identified by applying a 7-day temporal window to select closely successive heatwaves and extreme precipitation events. Subsequently, Equations (1)-(8) are used to calculate the attributes of CHWEP, single extreme precipitation, and single heatwave events. These attributes include:
- The number of occurrences of CHWEP and heatwave events,
- The average temperature intensity of CHWEP and single heatwave events,
- The average precipitation intensity of CHWEP and single extreme precipitation events,
- The average duration of heatwaves within CHWEP and single heatwave events.
    We thank the reviewer for pointing out this issue, which has helped us further refine and improve the quality of this study.

[Figure]

**Figure 1.** Schematic illustration of CHWEP and single extreme precipitation and heatwave events extracted from daily precipitation and daily maximum temperature time series over a year.

3. The authors do not explain how the ensembles are taken into account in their study. Are the above statistics computed on each member of the ensemble averaged out? Why is the number of members is different for the climate models than for the reanalysis? Does it pose a problem?

*Response:* In this study, to mitigate potential biases inherent in any single dataset, we adopted a multi-dataset approach by computing the mean across multiple datasets to reduce uncertainties associated with individual data sources. For the historical period (1980-2024), we used three reanalysis datasets, which are generally considered highly reliable and thus serve as benchmark datasets. For future (2056-2100) daily precipitation and daily maximum temperature data, we selected four CMIP6 global climate models that provide both variables. We first computed the ensemble mean of these four models and then applied the quantile delta mapping (QDM) method to bias-correct the model outputs. The CHWEP and single extreme events in the future period were subsequently identified using this bias-corrected, ensemble-mean data. It is worth noting that relatively few CMIP6 models provide daily maximum temperature (namely tasmax) outputs at the daily time scale, therefore, we carefully selected four models that simultaneously deliver both future daily precipitation and daily maximum temperature. We thank the reviewer for raising this point, which has helped us further refine the methodology and improve the clarity and accuracy of this manuscript.

4. The vocabulary is not consistent throughout the paper. For example, according to line 127, $F_C$ and $F_S$ denote frequencies (hence the $F$ letter), but Figure 2(a) and 2(b) show counts. Note also that according to caption, Figures 2c and 2d show heatwave intensity (between 0 and 40 °C). Why "intensity", and not "temperature"? Note also that according to Eqs. (3) and (4), $IT_C$ and $IT_S$ are means over sums, which I expect to be above 100°C?

*Response:* The authors have standardized the labels in both the text and figures. Figure 2a and b show the average annual number of CHWEP and single heatwave events, respectively, from 1980 to 2024, i.e., the frequency of these events per year. Similarly, Figure 2c and d represent the average intensity of CHWEP and single heatwave events per year over the same period (1980-2024), defined as the average daily temperature during heatwave events. Using the term "intensity" aligns better with conventional practice in this field of research. We thank the reviewer for raising this point, which has helped the authors improve the accuracy of the manuscript's presentation.

5. The test used in this study is the Wilcoxon rank-test (WRT), which is consistent under specific assumptions on the two distributions (say $X$ for a statistic computed in the past, and $Y$ for the same statistics computed in the future). The WRT is consistent if the alternative is that $Y$ is stochastically larger than $X$, i.e. $P(Y* > X*) \geq P(X* > Y*)$, where $X*$ and $Y*$ are random values from $X$ and $Y$, respectively. The test on the median (line 139) is consistent with the additional assumption of that alternative is restricted to a shift in location, i.e. $F_Y(s) = \delta + F_X(s)$, with $\delta > 0$.

In any case, it is misleading to state that "the WRST is emplyed to assess differences in extreme events characteristics" (line 135-136). The authors should reformulate and be more specific (and narrow).

*Response:* The authors have appropriately revised lines 135-136 in Section 3.3 to clarify that the Wilcoxon rank-sum test (WRST) is employed in this study to assess differences in the medians of extreme event characteristics. The WRST offers several advantages: it does not require the two samples to follow (or approximate) a normal distribution, and it imposes minimal requirements on sample size. In contrast, the t-test, used to evaluate whether the means of two groups differ significantly, requires either paired data (for the paired t-test) or independent samples that are approximately normally distributed, especially in small-sample settings, which limits its applicability. We thank the reviewer for highlighting this issue, as it has helped the authors further refine the study and improve the accuracy of the manuscript's presentation.

6. Section is almost impossible to follow; the authors compare quantities that have not been properly defined, such as standard deviations SD (line 242), "ratio of observed to random probability" (line 243). I could not make any sense of this.

*Response:* The authors have carefully revised these sections and added explanatory text regarding the concept of quantiles. In Section 3.4, we have included the definition of the probability ratio calculation formula and incorporated the use of standard deviation (SD) to characterize the variability of probability changes. Since quantiles and SD are commonly used statistical metrics in hydrometeorological studies, the authors did not provide extensive elaboration on them. We thank the reviewer for pointing out these issues, which have helped the authors further refine and improve the quality of this study.

Other comments:
Line 110: "The threshold for future periods is determined based on the threshold of historical period" is unclear to me. Does it mean they are equal? If not, what is formula to go from the threshold in the past to that in the futyre?

*Response:* We applied quantile delta mapping (QDM) to bias-correct the future precipitation and temperature data, ensuring that the thresholds for extreme precipitation and heatwave events, derived from reanalysis data during the historical period, can be consistently applied. In this study, we used the 90th percentile as the threshold to ensure a sufficient number of extreme events are captured. We have revised this sentence to improve clarity and accuracy. We thank the reviewer for highlighting this issue, which has helped the authors further refine and enhance the quality of this study.

Caption of Figure 1: "Identification ...events"

*Response:* The authors have revised Figure 1 to make its content more comprehensive and have also updated the figure caption. The authors thank the reviewer for pointing out this issue, as it helps us continuously improve the content and structure of the manuscript.

[Figure]

**Figure 1.** Schematic illustration of CHWEP and single extreme precipitation and heatwave events extracted from daily precipitation and daily maximum temperature time series over a year.

Line 167: One could argue that single heat waves are prevalent in all desert regions, where it only rains when it gets cooler

*Response:* As shown in Figure 2b, single heatwave events occur not only more frequently in desert regions but also exhibit high occurrence frequencies in tropical areas such as eastern Brazil and the Congo Basin, as well temperate regions including the U.S. West Coast and Central Asia. Furthermore, we employed a relative threshold to identify heatwave events in each region, which better captures the local characteristics of temperature variability. The authors have revised this sentence to ensure the description aligns more accurately with the results. We thank the reviewer for highlighting this issue, as it has helped the authors gain a deeper understanding of the significance and findings of this study.

Caption of all Figures: change "frequency" to "count"

*Response:* According to Equations 4 and 5 in Section 3.2, we calculated the frequencies of CHWEP and single extreme events in every year. Therefore, the authors have changed the label in all figures from "Count" to "Frequency," which better aligns with the terminology commonly used in hydrometeorological extreme event studies. We thank the reviewer for pointing out this issue, as it helps improve the accuracy of the manuscript's presentation.

Line 242, how are the standard deviations computed?

*Response:* The authors have already provided the formula for standard deviation (SD) in Equation 15 of Section 3.4. In this study, we calculated the SD of the observed encounter probability, the randomized encounter probability, and their ratio to quantify the interannual variability of CHWEP events. We thank the reviewer for raising this point, as it helps the authors continuously refine and enhance the content and structure of this study.

Caption of Figure 7: reference to panel (c) is missing

*Response:* The authors have added a reference to panel (c) in the caption of Figure 7. We have also carefully reviewed the entire manuscript to identify and correct similar oversights. The authors thank the reviewer for pointing out this omission, as it helps them continuously improve and refine this study.

---

## Author Comment (AC2)

* * *
**Response to the RC2**

Dear Reviewer,

We sincerely thank the reviewer for your continued interest in and support of our research. Your insightful comments and constructive suggestions have been immensely helpful in further refining and enhancing both the content and structure of this study. Upon receiving your review, we immediately revised the manuscript in accordance with your recommendations, making every effort to meet all the revision requirements. The modified sections have been highlighted in red in the manuscript for ease of review. Below, we provide our point-by-point responses to your comments.

Thank you for your time and kind assistance.

Best regards,
Haoyu Jin

This study analyzes historical and future changes in compound heatwave–extreme precipitation events (CHWEPs) and compares them with single extreme events at the global scale from 1980–2100. They used three reanalysis datasets as observations and four CMIP6 models under two SSP scenarios (SSP2-4.5 and SSP5-8.5). They found that CHWEPs exhibit higher frequency, stronger precipitation, and longer heatwave duration in mid-to-high latitudes, and that CHWEPs occurring in the tropics feature more intense heatwaves than single heatwave events. They also suggest that these patterns will persist in future projections. In addition, they found that although both CHWEPs and single extreme events increase across most of the globe in the future, extreme precipitation following heatwaves will become notably more intense and frequent.

I agree that this is a very important topic in the field, but I find the manuscript still quite far from the standard typically expected for HESS. The choices of observational and model data are not reasonably justified, the statistical techniques are not analyzed in a way that convinces me of the robustness of the results, and the results/discussion remain largely descriptive and shallow (not quantitatively or statistically convincing). Many of the figures repeat similar information without adding new insights, and some key quantities are not clearly defined. In my view, the authors will need extensive methodological revision, additional testing, and substantial rewriting before the paper can be considered for publication in HESS. Below I list some of my concerns for the authors to address (with a few general examples):

The rationale for choosing ERA5, MERRA-2, and JRA-55 for observations is unclear. JRA-55 is also an older dataset, and it has already been replaced by JRA-3Q.

*Response:* The authors have added an explanation for selecting these three reanalysis datasets. First, the ERA-Land post-processed daily statistics provide daily precipitation and daily maximum temperature data that have been widely validated in numerous studies and are known for their high accuracy. Building on this, we further incorporated MERRA-2 and JRA-55 datasets. By averaging

these three reanalysis datasets, we constructed a more robust dataset, which serves as the basis for identifying extreme precipitation and heatwave events, and subsequently detecting CHWEP events. The inclusion of JRA-55 was also motivated by its relatively smaller data volume while still meeting the study's requirements, in future work, we plan to adopt the newer JRA-3Q dataset. We thank the reviewer for raising this point, as it helps us further refine and enhance the quality of this study.

Line 91: It is unclear and confusing what "ensemble mean" for observational data refers to here. The three reanalysis datasets are deterministic and have different systematic biases, so combining them as an ensemble requires a more explicit explanation.

*Response:* The authors have carefully revised this sentence. In this study, we selected three commonly used reanalysis datasets. Considering that reanalysis data may contain biases, we used the mean of these three datasets as the most reliable estimate. Based on this averaged time series, we derived thresholds for extreme precipitation and heatwaves, which were then used to identify single extreme events and CHWEP events. Similar to most climate change studies, we employed a multi-dataset mean to minimize potential biases associated with any single dataset. We thank the reviewer for pointing out this issue, as it helps the authors continuously improve the content and structure of this study and enhance the clarity of the manuscript.

The paper uses only four GCMs from CMIP6 without explaining why they were selected over others.

*Response:* In this study, the four selected GCMs offer higher spatial resolution and greater accuracy compared to other GCMs. More importantly, they provide daily maximum temperature data for both the historical period and the future SSP2-4.5 and SSP5-8.5 scenarios. Since heatwave events in this study are identified based on daily maximum temperature, this capability is crucial, many other GCMs rarely offer daily-resolution maximum temperature datasets. The authors have already included an explanation at this location. We sincerely thank the reviewer for their valuable comments and suggestions, which help us continually improve the content and structure of this manuscript.

In Section 3.1 (Identification and characteristics of CHWEP events), when defining heatwaves and extreme precipitation events, it would help to mention the actual threshold values being used.

*Response:* The authors have already added a description in Section 3.1 regarding how the thresholds for heatwave and extreme precipitation events were determined. Indeed, defining these thresholds is crucial. In this study, the threshold for heatwave events is based on the 90th percentile of the historical (1980-2020) gridded daily maximum temperature, while the threshold for extreme precipitation events is derived from the 90th percentile of the gridded precipitation data on wet days (>1 mm) during the same historical period. The 90th percentile was chosen to ensure a sufficient number of heatwave and extreme precipitation events for analysis. The authors appreciate the reviewer's comment on this point, as it helps them further refine and improve the study.

The definitions of formulas (1)–(8) are not very clear. Some metrics are calculated across all events while others are computed per event, and this is confusing. These formulas are difficult to interpret and need clearer and more accurate notation and description.

*Response:* The authors have revised the equations in Section 3.2 to enhance their clarity and added annotations for the relevant variables in Figure 1. Identifying single extreme events, as well as heatwaves, extreme precipitation events, and CHWEP (compound heatwave and extreme precipitation) events, is central to this study. Although these concepts are straightforward and intuitively understandable, their practical implementation demands substantial computational resources. In this study, for instance, analyzing single extreme events and CHWEP over the historical period (1980-2024) on a computer equipped with 8 GB of RAM and a 1.4 GHz Quad-Core Intel Core i5 processor required approximately nine hours. Given the differences among single extreme events and CHWEP events, all attributes of these events (except for occurrence frequency and duration) were averaged on a per-day basis in this study to facilitate comparison. We thank the reviewer for raising this point, which has helped the authors further refine and improve the work.

[Figure]

**Figure 1.** Schematic illustration of CHWEP and single extreme precipitation and heatwave events extracted from daily precipitation and daily maximum temperature time series over a year.

Some notations and units in the figures are not clear. For example, in Fig. 2a/b, what is the meaning of the units of "time"? Do you mean the total number of events over 1980–2024 or the number of events per year?

*Response:* The authors have revised the notations in Figures 2a and 2b to clarify that they represent the average annual number, namely the average frequency of single heatwave events and CHWEP events during the historical period (1980-2024). We thank the reviewer for pointing out this issue, which has helped the authors further refine the manuscript and enhance the accuracy of its presentation.

Some discussions of the results are insufficient or inaccurate. For example, in the discussion of Fig. 3c at line 181 ("Across most mid-to-high latitude areas, the extreme precipitation intensity during CHWEPs exceeds that of single extreme precipitation events"), the authors only focus on the Northern Hemisphere and ignore the Southern Hemisphere mid-high latitudes.

*Response:* The authors have carefully revised this statement to clarify that the significant regions are primarily concentrated in the mid- to high-latitude areas of the Northern Hemisphere. Although some parts of the mid- to high-latitude Southern Hemisphere also exhibit stronger precipitation in CHWEP events compared to single extreme precipitation events, the signal is not as pronounced. We thank the reviewer for highlighting this issue, which has helped the authors improve the accuracy of the study's presentation.

Much of the Results section is purely descriptive. The analysis could be more quantitative, for example by providing regional averages or more meaningful values to make the study deeper and more informative.

*Response:* The authors have thoroughly revised the Results section, providing a more in-depth description of the findings and expanding the comparison of regional differences to better understand how single extreme events, including heatwaves and extreme precipitation events, as well as CHWEP events, vary across regions. Overall, by comparing CHWEP events with single extreme events, this study has uncovered several intriguing phenomena: for instance, extreme precipitation following heatwaves tends to be more intense than precipitation from isolated extreme precipitation events; the mid- to high-latitude regions of the Northern Hemisphere emerge as hotspots for CHWEP occurrence; and extreme precipitation is more likely to occur after heatwave events. We thank the reviewer for raising this point, which has helped the authors further refine and enhance the quality of this study.

Some figures need to be modified and improved. For example, the colors for "Future SSP2-4.5 single" and "Future SSP5-8.5 single" in Fig. 5 are too similar to distinguish clearly.

*Response:* The authors have modified the line colors in Figure 5 to enhance their distinguishability. Similarly, the line colors in Figure 6 have also been adjusted for improved clarity. Furthermore, the authors have reviewed all figures throughout the manuscript and made selective color adjustments to enhance the visual clarity and prominence of the results. We thank the reviewer for pointing out these issues, which have helped the authors continuously improve the quality of the figures in this paper.

[Figure]

**Figure 5.** Changes in the frequency (a), heatwave intensity (b), extreme precipitation intensity (c), and heatwave duration (d) during the historical and future periods of CHWEP and single extreme events.

[Figure]

**Figure 6.** Comparison of the cumulative probability of the median differences between the frequency (a), heatwave intensity (b), extreme precipitation intensity (c), and heatwave duration (d) of CHWEP and single extreme events under the SSP2-4.5 and SSP5-8.5 scenarios and the historical period.

Although the authors introduce the Wilcoxon rank-sum test and event coincidence analysis in their Methods section, no p-values or other statistical results are shown. Without reporting these outcomes, the reader cannot assess whether the stated differences or assumptions are statistically meaningful.

*Response:* The authors have marked the regions where the median differences in the sequences obtained through the Wilcoxon rank-sum test (WRST) are statistically significant, and have further calculated the proportions of non-significant positive (NSP), significant positive (SP), non-significant negative (NSN), and significant negative (SN) results. We thank the reviewer for raising this point, which has helped the authors further refine and enhance the quality of this study.

---

## Author Comment (AC3)

* * *
**Response to the RC3**

Dear Reviewer,

We sincerely thank the reviewer for your care and support of our research. Your valuable comments and suggestions have been of tremendous help in continuously refining and improving this study. Upon receiving the review comments, we promptly revised the manuscript in accordance with your suggestions, striving to fully address all revision requirements. The revised portions are highlighted in red in the manuscript for your convenience. Our point-by-point responses to your comments are provided below.

Thank you for your time and kind assistance.

Best regards,
Haoyu Jin

General comments

This study explores historical and future changes in the characteristics of compound heatwave-extreme precipitation events and the statistical linkage between heatwave and extreme precipitation using global reanalyses products and CMIP6 model projections. This topic is relevant given rising climate-driven extremes and the increasing importance of understanding compound hazards. The manuscript is clearly structured and the analyses are potentially valuable. However, in its current form, the analytical framework remains somewhat superficial. The study would benefit from more explicit justification of key methodological decisions, clearer articulation of its scientific novelty, and additional explanation to strengthen transparency and interpretability. Several assumptions are stated but not sufficiently supported, and important methodological steps lack the detail needed to ensure reproducibility. Addressing these issues would substantially improve the clarity, credibility, and overall impact of the work. With these enhancements, the study could make a meaningful contribution to understanding CHWEP events, but substantial conceptual, methodological, and interpretive clarifications are required before the conclusions can be fully supported and before the manuscript meets the standards expected for publication in HESS.

Specific comments
1. Your description of preprocessing of datasets is straightforward. However, I am concerned about the motivation for using ensemble mean as the final reference datasets. Given the uncertainties inherent in global reanalyses, the rationale for preferring ensemble mean over the best-performing reanalyses should be well articulated. Similarly, what is the justification for selecting four specific CMIP6 models? Is this based on their transient climate responses to avoid 'Hot Model' problems associated with CMIP6 (e.g. Hausfather et al., 2022)?

Reference

Hausfather, Z., Marvel, K., Schmidt, G.A., Nielsen-Gammon, J.W., Zelinka, M., 2022. Climate simulations: recognize the 'hot model' problem. Nature 605, 26–29. https://doi.org/10.1038/d41586-022-01192-2

*Response:* The authors selected multiple models and used the ensemble mean as the final time series for identifying extreme precipitation and heatwave events to enhance robustness. Reanalysis datasets, which assimilate multiple data sources, particularly in situ station observations, offer relatively high accuracy; therefore, we chose three commonly used reanalysis datasets and computed their mean to obtain a more reliable and stable result. The four CMIP6 GCMs were selected because they exhibit high accuracy, high spatial resolution, and provide daily maximum temperature data, essential for heatwave detection in this study. Moreover, as the reviewer noted, this approach helps avoid bias introduced by so-called "hot models" that may overestimate warming. We thank the reviewer for raising this point, which has deepened our understanding of best practices in GCM selection and application.

2. The quantile mapping approach is mentioned only superficially, it is unclear whether temperature and precipitation were corrected separately, future projections were adjusted using historical distributions, and potential risks of overcorrection, particularly for extremes?. Additionally, since CHWEP events potentially rely on joint behavior of temperature and precipitation, it might be helpful if the authors note whether the bias correction approach preserves multivariate dependence and temporal structure. Quantile mapping can distort these if applied independently.

*Response:* The authors applied Quantile Delta Mapping (QDM) to bias-correct precipitation and temperature projections for the future period. The bias-correction formula for precipitation is given in Equation (2) of Section 3.1, and that for temperature in Equation (3) of the same section. Because QDM preserves the relative structure of the data, it does not alter the spatial or temporal occurrence patterns of heatwaves or extreme precipitation events. In this study, high data accuracy is essential when comparing single extreme events and CHWEP events between the future and historical periods. However, when comparing single extreme events with CHWEP events within the same period, absolute data accuracy is less critical, as the analysis focuses on the differences between the two event types rather than their absolute magnitudes. We thank the reviewer for raising this point, which has helped the authors further refine and improve the study.

3. "Historical daily maximum temperatures" is vague. Please clarify the baseline climate period used to calculate threshold, and whether the chosen period affects the comparability between historical and future events?. The authors should add a bit more context (e.g. land-atmosphere memory, soil moisture decay timescales) to support the choice of 7-day window. Also clarify whether intensities represent mean values per event or aggregated means across all events.

*Response:* Indeed, the selection of thresholds for extreme precipitation and heatwave events is critical to their identification. In this study, the heatwave threshold at each grid cell is defined as the 90th percentile of the 45-year (1980-2024) local daily maximum temperature series during the historical period. Similarly, the threshold for extreme precipitation is set as the 90th percentile of the wet-day (>1 mm) precipitation series over the same 45-year period at each grid point. The 90th percentile is chosen to ensure a sufficient number of extreme events for robust statistical analysis.

A 7-day window is adopted to balance the need for physically plausible linkage between heatwaves and subsequent extreme precipitation while maintaining an adequate sample size; this window length follows the approach used in You and Wang's study. In Section 3.2, we have revised the formulations of all relevant indices to enhance clarity. Specifically, we compute the annual mean of all events of a given type, ensuring that each year contributes only one averaged value per extreme-event metric. We thank the reviewer for raising these points, which have helped us further refine and improve the quality of this study.

[Figure]

**Figure 1.** Schematic illustration of CHWEP and single extreme precipitation and heatwave events extracted from daily precipitation and daily maximum temperature time series over a year.

4. Please clarify whether ECA uses events derived from the earlier definitions or whether counts are based on individual days. ECA may be sensitive to the way the underlying event is defined. In addition, consider clarifying ECA only tests for statistical dependence, not physical causality. The text currently states it can "identify possible causal relationships,.." which may be an overstatement.

*Response:* In this study, we calculated two probabilities based on the ECA algorithm: one is the observed co-occurrence probability of heatwave and extreme precipitation events within a 7-day time window, and the other is the random co-occurrence probability obtained by randomly shuffling the timing of heatwave and extreme precipitation events across a year and then calculating the likelihood of their co-occurrence within a 7-day window. Both extreme precipitation events and heatwaves were identified using thresholds specific to each event type. The authors have carefully revised this sentence to ensure more accurate wording and to avoid overstating the role of the ECA algorithm. We thank the reviewer for pointing out this issue, which has helped us improve the precision of our presentation.

5. A brief mechanistic explanation why some patterns (e.g. "hotspots in the Sahara Desert, the Middle East, and Australia", "CHWEPs are more pronounced in mid-to-high latitudes")  emerge or what they imply would strengthen the narrative. Statements such as "CHWEPs occur more

frequently" or "intensity is significantly higher" would benefit from at least approximate magnitudes (e.g., percent differences)

*Response:* The authors suggest that the stronger extreme precipitation following heatwaves in the mid- to high-latitude regions of the Northern Hemisphere, as well as the higher frequency of isolated heatwave events in tropical and desert regions, may be attributed to the fact that rising temperatures in the mid- to high-latitude Northern Hemisphere enhance the hydrological cycle, thereby promoting extreme precipitation, whereas tropical and desert regions experience distinct wet and dry seasons, leading to more frequent and longer-lasting individual heatwave events. To further investigate the spatiotemporal distribution differences between isolated extreme events and CHWEPs, the authors additionally extracted regional extreme event attribute values and their ratios. The authors thank the reviewer for raising this point, which helps them continuously improve and refine this study.

6. The statement that "CHWEP precipitation intensity under SSP2-4.5 exceeds single-event intensity under SSP5-8.5" is interesting, please provide a short explanation e.g. heatwave-induced atmospheric instability?

*Response:* One of the key findings of this study is that more intense extreme precipitation tends to occur following heatwaves. Heatwave events accelerate moisture evaporation and enhance the hydrological cycle, particularly in the mid- to high-latitude regions of the Northern Hemisphere, where the climate is generally colder. The heatwaves in these regions boost evaporation, increasing atmospheric instability and thereby promoting precipitation. We gratefully acknowledge the reviewer for your valuable comments and suggestions, which have greatly helped the authors further refine and improve this study.

7. The authors mentioned widely known ideas (e.g., compound events cause more damage than single events) without tying them closely to your specific results or demonstrating how your findings confirm, extend, or challenge this existing knowledge. Also you state that integrated intensities exceed those of single events, but do not explain what this means physically (e.g. more moisture availability? stronger thermal anomalies?). A more explicit comparison would help interpret the significance of CHWEP intensification.

*Response:* The authors have revised the relevant sections, placing greater emphasis on the significant contribution of this study to understanding CHWEP events. The study consists of three main components. First, using daily precipitation and daily maximum temperature data from the historical period (1980-2024), we identified individual extreme events and CHWEP events, and further analyzed their differences. We found in the mid- to high-latitude regions of the Northern Hemisphere, CHWEP events occur more frequently and with greater precipitation intensity compared to individual extreme events. Second, applying future daily precipitation and daily maximum temperature data (2056-2100), we extracted future individual extreme events and CHWEP events and observed similar spatial distribution patterns. Third, we employed the ECA method to compare the actual probability of extreme precipitation following a heatwave with the probability expected under random co-occurrence. The results indicate that extreme precipitation events are significantly more likely to occur after heatwaves than by chance alone. The authors propose that heatwave events intensify the hydrological cycle and enhance atmospheric instability,

thereby triggering extreme precipitation, particularly in the mid- to high-latitude regions of the Northern Hemisphere. We sincerely thank the reviewer for your insightful comments and constructive suggestions, which have greatly assisted us in further refining and improving this study.

8. '..both CHWEP and single events are increasing significantly in frequency, intensity, and duration', it is unclear which regions or magnitude of change? More details should be added to make the discussion more insightful.

*Response:* The authors have added a description in the Discussion section regarding the regions where extreme events are projected to intensify in the future. Specifically, CHWEP and individual heatwave events are expected to increase more in frequency in equatorial regions; CHWEP and individual heatwaves will exhibit greater increases in temperature intensity in the mid- to high-latitude regions of the Northern Hemisphere; CHWEP and individual extreme precipitation events will show larger increases in precipitation intensity in equatorial regions; and the duration of CHWEP and individual heatwave events will lengthen more significantly over the southwestern United States, the Sahara Desert, and India. The authors further extracted and compared metrics of CHWEP and single extreme events across different regions, presenting box plots in the supplementary materials to provide a clearer comparison of regional changes. We appreciate the reviewer for pointing out this issue, as it has helped the authors further refine and enhance the quality of this study.

[Figure]

**Figure S1.** Regional division based on continental boundaries.

[Figure]

**Figure S2.** Box plots of the frequency (a), heatwave intensity (b), extreme precipitation intensity (c), and heatwave duration (f) for CHWEP and single extreme events.

9. '..but also reveals a dangerous "rapid transition" phenomenon…' This is one novel contribution that should be elaborated (currently the discussion is brief).

*Response:* The authors have added a more detail description of rapid-transition extreme events in the Discussion section. "Rapid transition from heatwave to extreme precipitation" refers to the phenomenon in which a prolonged heatwave abruptly shifts to intense or even extreme precipitation within a short period, typically several hours to a few days. Such events exemplify the dynamic connections and rapid evolution between distinct types of extreme weather in the climate system. They have been increasingly observed in many parts of the world in recent years, with particularly pronounced occurrences in mid- to high-latitude regions. The plausible physical mechanisms include: intense surface heating during heatwaves, which increases atmospheric instability, and enhanced evapotranspiration, which supplies additional moisture to the atmosphere. When favorable large-scale circulation conditions, such as cold-air intrusion or low-level jets emerge, the accumulated energy and moisture are rapidly released, triggering strong convection and extreme precipitation. This rapid "dry-to-wet" transition between contrasting extremes not only amplifies compound disaster risks, such as flash flooding following heat stress, but also poses heightened challenges for climate forecasting, early-warning systems, and emergency response frameworks. The rapid interconversion between different types of extreme events has become a focal topic in recent hydrometeorological research (Chen et al., 2022; Tan et al., 2023; Woolway

et al., 2021). We sincerely thank the reviewer for your valuable comments and suggestions, which have greatly helped the authors further refine and improve this study.

Chen, Y., Liao, Z., Shi, Y., Li, P., and Zhai, P.: Greater Flash Flood Risks From Hourly Precipitation Extremes Preconditioned by Heatwaves in the Yangtze River Valley, Geophys. Res. Lett., 49, https://doi.org/10.1029/2022GL099485, 2022.
Tan, X., Wu, X., Huang, Z., Fu, J., Tan, X., Deng, S., Liu, Y., Gan, T. Y., and Liu, B.: Increasing global precipitation whiplash due to anthropogenic greenhouse gas emissions, Nat. Commun., 14, 2796, https://doi.org/10.1038/s41467-023-38510-9, 2023.
Woolway, R. I., Kraemer, B. M., Zscheischler, J., and Albergel, C.: Compound Hot Temperature and High Chlorophyll Extreme Events in Global Lakes, Environ. Res. Lett., 16, 124066, https://doi.org/10.1088/1748-9326/ac3d5a, 2021.

10. This section is missing the acknowledgment of potential limitations of the methods/results (e.g. uncertainties in climate data, bias correction, sensitivity to thresholds). This is important to strengthen credibility and transparency.

*Response:* The authors have added a discussion in the Discussion section regarding potential limitations of this study and aspects that warrant further improvement. In the present study, we enhanced the robustness of our results through careful experimental design, for example, by using the ensemble mean of outputs from multiple models, defining extreme events using the 90th percentile as the threshold, and applying a 7-day time window to identify sequences of compound events, in order to obtain a sufficient number of extreme event samples. In future work, it would be valuable to include a broader range of climate models, test alternative percentile thresholds for defining extremes, and explore different time windows between successive extreme events to further verify the reliability and accuracy of our findings. We sincerely thank the reviewer for your insightful comments and valuable suggestions, which have greatly helped the authors to continuously improve and refine this study.

Minor comments
L50-55: It will be useful to connect which category CHWEP belongs to (e.g. temporally compounding). This will help readers who may not be familiar with the terminology.

*Response:* We have already clarified at this point that our study primarily focuses on heatwaves and extreme precipitation events that are temporally consecutive, namely, temporally linked CHWEP events. We thank the reviewer for raising this issue, as it has helped the authors further refine and improve the quality of this study.

L70-75: I would be careful with the use of the phrase ' the first comprehensive, global scale'. Ensure your phrase is defensible or rephrase to avoid overclaiming.

*Response:* The authors have appropriately revised this sentence to achieve a more balanced expression. An important improvement of this study compared to previous work is the systematic comparative analysis of the spatiotemporal distribution differences between CHWEP events and single extreme events, with greater emphasis placed on the distinct regional patterns exhibited by

CHWEP events relative to single extreme events. We thank the reviewer for pointing out this issue, which has helped the authors further refine and enhance the quality of this study.

L75-80: The authors mentioned 'statistical significance of differences among event types', however, it is unclear whether this pertains to frequency, duration, temporal lag, etc.?. As I mentioned previously, the authors should consider clarifying that CHWEP events are defined both in terms of temporal thresholds and sequence, if that is indeed the approach.

*Response:* The authors have appropriately revised this paragraph. CHWEP events are composed of consecutive heatwaves and extreme precipitation events, with a time window of no more than 7 days. Single heatwave events and single extreme precipitation events each have their own respective metrics. We compare the heatwave metrics within CHWEP events against those of single heatwaves, and similarly compare the extreme precipitation metrics within CHWEP events against those of single extreme precipitation events. This allows us to isolate the distinctive characteristics of CHWEP events relative to single extreme events, thereby enabling an evaluation of their spatiotemporal differences. We thank the reviewer for your valuable comments and suggestions, which help the authors continuously refine and improve this study.

L85: Use standard citations instead of hyperlinks throughout the texts. Links should be provided in the appropriate 'data availability' section

*Response:* The authors have removed the direct links to these data and instead provided appropriate references. Links to the data are included in the Data Availability section. We thank the reviewer for highlighting this issue, which has helped the authors further improve the content and structure of the manuscript.

Fig 1: Caption is too short. Consider adding more information.

*Response:* The authors have revised and redrawn Figure 1 and updated its caption accordingly. We thank the reviewer for pointing out this issue, which has helped the authors continuously refine and enhance this study.

[Figure]

**Figure 1.** Schematic illustration of CHWEP and single extreme precipitation and heatwave events extracted from daily precipitation and daily maximum temperature time series over a year.

---

## Author Comment (AC4)

* * *
**Response to the CC1**

Dear Reviewer,

We sincerely thank the reviewer for your support and insightful feedback, which greatly improved our study. We have carefully revised the manuscript in response to your comments, highlighted in red, and provide point-by-point replies below.

Thank you for your time and kind assistance.

Best regards,
Haoyu Jin

CC1

The method of this study raises some questions for me. They use three reanalysis datasets (ERA5, MERRA-2, JRA-55) as historical data and use four CMIP6 models under two SSP scenarios for the future. Firstly, they address the systematic biases between climate model simulations and reanalysis data. They highlight that they performed a simple bias correction on the CMIP6 model output although, they note that, since they are comparing CHWEP events and single extreme events, the precise accuracy of the absolute value is not important for the conclusion. However, this is too short of an explanation in my opinion. Limitations of the bias correction method and explaining it shortly would be more complete.

*Response:* The authors have added in Section 3.1 a description of how future climate model data were bias-corrected in this study, employing the Quantile Delta Mapping (QDM) algorithm, which preserves the trend characteristics of the original time series. Regarding the comparison between CHWEP events and single extreme events in both historical and future periods, since both types of events are derived from the same time series, their relative differences remain valid regardless of potential uncertainties in the absolute data accuracy, thus ensuring the robustness of the conclusions drawn. Furthermore, the use of a multi-model mean, a 90th percentile threshold, and a 7-day temporal window were all implemented to yield robust results and enhance the reliability of the findings. We sincerely thank the reviewer for your valuable comments and suggestions, which have greatly contributed to the continuous improvement and refinement of this study!